# Histone demethylase JMJD2D protects against enteric bacterial infection via up-regulating colonic IL-17F to induce β-defensin expression

Yong Zhang[1,2☯], Bei Li[1☯], Yilin Hong[1☯], Ping Luo[3], Zaifa Hong[4], Xiaochun Xia[5], Pingli Mo[1], Chundong Yu[1]*, Wenbo Chen[3]*

**1** State Key Laboratory of Cellular Stress Biology, Innovation Center for Cell Biology, School of Life Sciences, Xiamen University, Xiamen, China, **2** Department of Pathology, Fujian Medical University Union Hospital, Fuzhou, China, **3** Department of Cardiology, Xiamen Key Laboratory of Cardiac Electrophysiology, Xiamen Institute of Cardiovascular Diseases, The First Affiliated Hospital of Xiamen University, School of Medicine, Xiamen University, Xiamen, China, **4** Department of Hepato-Biliary-Pancreatic and Vascular Surgery, The First Affiliated Hospital of Xiamen University, School of Medicine, Xiamen University, Xiamen, China, **5** Xiamen Medical College, Xiamen, China

☯ These authors contributed equally to this work.
* cdyu@xmu.edu.cn (CY); enwinbob@163.com (WC)

**Data Availability Statement:** The authors confirm that all data underlying the findings are fully available without restriction. All relevant data are

## Abstract

Histone demethylase JMJD2D (also known as KDM4D) can specifically demethylate H3K9me2/3 to activate its target gene expression. Our previous study has demonstrated that JMJD2D can protect intestine from dextran sulfate sodium (DSS)-induced colitis by activating Hedgehog signaling; however, its involvement in host defense against enteric attaching and effacing bacterial infection remains unclear. The present study was aimed to investigate the role of JMJD2D in host defense against enteric bacteria and its underlying mechanisms. The enteric pathogen *Citrobacter rodentium* (*C. rodentium*) model was used to mimic clinical colonic infection. The responses of wild-type and JMJD2D[-/-] mice to oral infection of *C. rodentium* were investigated. Bone marrow chimeric mice were infected with *C. rodentium*. JMJD2D expression was knocked down in CMT93 cells by using small hairpin RNAs, and Western blot and real-time PCR assays were performed in these cells. The relationship between JMJD2D and STAT3 was studied by co-immunoprecipitation and chromatin immunoprecipitation. JMJD2D was significantly up-regulated in colonic epithelial cells of mice in response to *Citrobacter rodentium* infection. JMJD2D[-/-] mice displayed an impaired clearance of *C. rodentium*, more body weight loss, and more severe colonic tissue pathology compared with wild-type mice. JMJD2D[-/-] mice exhibited an impaired expression of IL-17F in the colonic epithelial cells, which restricts *C. rodentium* infection by inducing the expression of antimicrobial peptides. Accordingly, JMJD2D[-/-] mice showed a decreased expression of β-defensin-1, β-defensin-3, and β-defensin-4 in the colonic epithelial cells. Mechanistically, JMJD2D activated STAT3 signaling by inducing STAT3 phosphorylation and cooperated with STAT3 to induce IL-17F expression by interacting with STAT3 and been recruited to the IL-17F promoter to demethylate H3K9me3. Our study demonstrates that JMJD2D contributes to host defense against enteric bacteria through up-regulating IL-17F to induce β-defensin expression.

within the paper and its Supporting Information files.

**Funding:** We acknowledge funding by the Natural Science Foundation of Fujian Province (2021J05289 to WC, 2022J011404 to XX and 2023J011621 to ZH), the National Natural Science Foundation of China (81970485, 82173086 and 82371764 to CY), the Science and Technology Project of Xiamen Medical College (K2022-04 to XX). The funders had no role in study design, data collection and analysis, decision to publish, or preparation of the manuscript.

**Competing interests:** The authors have declared that no competing interests exist.

## Author summary

Our study aimed to unravel the role of JMJD2D, a histone demethylase also known as KDM4D, in the host's defense against enteric bacterial infections and explore the underlying mechanisms. While we had previously demonstrated JMJD2D's protective effect in DSS-induced colitis by activating Hedgehog signaling, its involvement in defending against enteric attaching and effacing bacterial infections remained unexplored. To address this, we employed a *Citrobacter rodentium* (*C. rodentium*) infection model to mimic colonic infections in a clinical context. In response to *C. rodentium* infection, there was a notable increase in JMJD2D expression within the colonic epithelial cells of mice. Our investigation compared the responses of wild-type mice with those lacking JMJD2D (JMJD2D$^{-/-}$) following oral *C. rodentium* infection. Notably, JMJD2D$^{-/-}$ mice exhibited impaired *C. rodentium* clearance, greater body weight loss, and more severe colonic tissue damage compared to their wild-type counterparts. Subsequent analysis revealed that JMJD2D deficiency resulted in reduced IL-17F expression in colonic epithelial cells. IL-17F is a known regulator that restricts *C. rodentium* infection by promoting the expression of antimicrobial peptides. Consequently, JMJD2D$^{-/-}$ mice showed diminished expression of critical antimicrobial peptides, including β-defensin-1, β-defensin-3, and β-defensin-4, in their colonic epithelial cells. Mechanistically, we discovered that JMJD2D played a pivotal role in activating STAT3 signaling by enhancing STAT3 phosphorylation. Moreover, JMJD2D collaborated with STAT3 to induce IL-17F expression. This cooperation involved JMJD2D physically interacting with STAT3 and binding to the IL-17F promoter, where it demethylated H3K9me3. In summary, our research reveals that JMJD2D contributes significantly to the host's defense against enteric bacterial infections by up-regulating IL-17F, which in turn induces the expression of antimicrobial peptides such as β-defensins. Our findings provide valuable insights into the molecular mechanisms governing host defense against enteric bacterial infections, emphasizing the potential therapeutic relevance of targeting JMJD2D in such contexts.

## Introduction

Infection of human-specific gastrointestinal pathogens including enteropathogenic *Escherichia coli* (EPEC) and enterohaemorrhagic *E. coli* (EHEC) is a leading cause of diarrhoeal illness and mortality worldwide [1,2]. EPEC and EHEC form attaching and effacing (A/E) lesions at the intestinal mucosa surface, followed by colonizing the intestinal epithelial cells [3,4]. *Citrobacter rodentium* is a natural mouse-specific A/E enteric pathogen, which shares virulence factors and an infection strategy with human A/E enteric pathogens EPEC and EHEC [5]. Therefore, the model of *C. rodentium* infection of mice is widely used to study pathogenesis, host innate and adaptive immune responses to infection, and pathogen-host-microbiota interactions and immunometabolism [5,6].

In response to *C. rodentium* infection, the intestine of mouse produces many cytokines to induce antimicrobial peptides to protect host [7–9]. Among them, IL-17A and IL-17F play an important role. It has been reported that IL-17A and IL-17F protect mice against *C. rodentium* infection via the induction of inflammatory mediators and β-defensins [8]. Some studies have suggested that IL-17A is important for host defense against *C. rodentium* infection at the early stage, while IL-17F is more critical for controlling *C. rodentium* infection at the late stage [10]. IL-17A and IL-17F belong to IL-17 family that is composed of six members, including IL-17A,

IL-17B, IL-17C, IL-17D, IL-17E, and IL-17F. IL-17A shares 50% homology with IL-17F [11]. IL-17A is primarily produced by T cells, while IL-17F is produced by T cells, innate immune cells, and epithelial cells [8]. IL-17A and IL-17F can form homodimers and heterodimers to induce signal transduction through binding to the obligate IL-17RA-IL-17RC heterodimeric receptor complex [12,13], implying that they have similar signaling pathway and biological functions. Consistently, IL-17A and IL-17F enhance the induction of chemokines (such as CXCL1, CXCL2, CXCL5 and CCL2), cytokines (such as GM-CSF and IL-6), and antimicrobial peptides (such as β-defensin) via IL-17RA-IL-17RC-ACT1-TRAF6 signaling pathway in auto-immune diseases models and host-defense systems (such as *Staphylococcus aureus* and *Citrobacter rodentium*) [8,11,14].

Histone lysine demethylase JMJD2D, which belongs to the JMJD2 family, can demethylate lysine 9 on histone 3(H3K9) me2/me3 to regulate its target genes expression [15]. Our previous studies have shown that JMJD2D can promote colorectal cancer progression by enhancing Wnt/β-catenin, Hedgehog, and HIF1 signaling pathways [16–18], and protect intestine from DSS-induced colitis by activating Hedgehog signaling [17], however, its role in *C. rodentium*-induced colitis remains unclear.

In this study, we found that JMJD2D was up-regulated in the colonic epithelial cells after *C. rodentium* infection, suggesting that JMJD2D is involved in *C. rodentium*-induced colitis. We also found that JMJD2D$^{-/-}$ mice were more sensitive to *C. rodentium* infection compared to wild-type mice. JMJD2D$^{-/-}$ mice display an impaired expression of IL-17F in the colonic epithelial cells, which contributes to eliminate *C. rodentium* via increasing antimicrobial peptide expression. Furthermore, JMJD2D$^{-/-}$ mice exhibited a reduced expression of β-defensin 1, β-defensin 3, and β-defensin 4 in the colonic epithelial cells. Mechanistically, JMJD2D cooperated with STAT3 to enhance IL-17F expression to subsequently induce β-defensin expression.

## Materials and methods

### Ethics statement

The animal experimental design was approved by the Ethics Committee of the Ethics Committee of Xiamen University (approval number: XMULAC20190040). All experiments were performed in accordance with approved guidelines and related regulations.

### Mice

JMJD2D$^{-/-}$ mice on a C57BL/6 background were generated as described previously [16]. Wild-type mice were used as controls. Six-week-old male mice were used in all experiments. All experimental procedures were performed in accordance with animal protocols approved by the Laboratory Animal Center of Xiamen University.

### *C. rodentium* infection of mice and quantification of the bacterial burden

*C. rodentium* strain ATCC51459 was cultured into Luria-Bertani (LB) medium for 12 h at 37°C and 180 rpm shaking. Mice were administrated with $1.0 \times 10^9$ CFU *C. rodentium* in 0.2 ml PBS by oral gavage after fasting for 8 h. The number of live bacteria was determined by serially diluting and spreading on a MacConkey agar plate for 18–24 h at 37°C. For determination of *C. rodentium* in feces and tissue samples, feces, colons, feces + cecums, livers, spleens, and lungs were collected, weighed, immersed with 1ml PBS and homogenized in a Tissuelyser-24 for a total of 6 min at 30 Hz at room temperature. Feces and tissue homogenates were serially diluted in PBS and plated in triplicate onto MacConkey agar plates. *C. rodentium* colonies were identified as pink colonies and counted after an incubation time of 20 h at 37°C.

## Colon culture

The 0.5 cm distal colon was dissected and washed three times with cold PBS containing streptomycin and penicillin to removed feces. The colon segments were cultured in 1 mL RPMI 1640 medium without FBS but with streptomycin and penicillin for 24 h at 37˚C with 5% $CO_2$. Supernatants were harvested at 13,000×g for 10 min at 4˚C. And the supernatants were frozen in -80˚C for further analysis.

## Bone marrow chimeric mice

A single lethally irradiated (9 Gry) eight-week-old recipient wild-type mice and JMJD2D$^{-/-}$ mice were reconstituted with 1× $10^7$ femoral bone marrow cells of either wild-type mice or JMJD2D$^{-/-}$ mice via intravenous injection 4 h later after irradiation. The chimeric mice were divided into four groups: WT to WT, WT to JMJD2D$^{-/-}$, JMJD2D$^{-/-}$ to WT, JMJD2D$^{-/-}$ to JMJD2D$^{-/-}$ mice. Reconstituted mice were administered with antibiotics in the drinking water for 3 weeks during the recovery phase. The efficiency of bone marrow transplantation was determined after 4 weeks via digital PCR using peripheral blood genomic DNA and the averaged efficiency of bone marrow transplantation was calculated to exceed 90% (**S1 Fig**). Eight weeks after bone marrow transplantation, the chimeric mice were infected with *C. rodentium* and sacrificed on day 7 after *C. rodentium* infection.

## Cytokines measure

The concentrations of IL-6, IL-1β, and TNF-α in the colon culture supernatants were quantified by Ready-SET-Go ELISA sets (eBioscience) according to manufacturer's instructions.

## Histopathology

The distal colon tissues were removed, fixed in 10% neutral formalin and embedded in paraffin. Five-μm sections were prepared and stained with hematoxylin and eosin (H&E). Histological scoring for inflammation and tissue damage was determined as described previously [19].

## Immunohistochemistry

Five-micrometer paraffin sections were cut, deparaffinized and rehydrated. Antigens were retrieved by soaking in preheated citrate buffer (pH 6) under microwave heating for 20 min. Endogenous peroxidases were quenched by incubating with 3% $H_2O_2$ for 15 min. Nonspecific binding sites were blocked with 5% BSA for 1 h at room temperature. And then the sections were incubated overnight with rabbit anti-*C. rodentium* antibody at 4˚C. The next day, sections were washed and incubated with EliVision plus kits (Maixin) at room temperature. Peroxidase activity was assessed by DAB reagents.

## Immunofluorescence staining

Tissue section preparation and antigen repair were consistent with oboved immunohistochemistry assay. Nonspecific binding sites were blocked with 5% BSA for 1 h at room temperature. And then the sections were incubated overnight with rabbit anti-Ki67 (ab15580, Abcam) antibody at 4˚C overnight. Then sections were washed and incubated with Goat anti-Rabbit IgG (H+L) Alexa Fluor Plus 555 (A32732, Invirogen) at room temperature for 1h. Sections were washed with PBS for three times, and mounted with Solarbio Mounting medium with DAPI (S2110). Images were acquired using Zeiss AxioScan7.

## Isolation of colonic epithelial cells

The colonic epithelial cells were isolated from the colons using a modification of a previously described protocol [19].

## Cell lines

The murine colorectal cancer cell line CMT93, and human embryonic kidney cell line HEK293T were obtained from the Cell Bank of Type Culture Collection of the Chinese Academy of Sciences (Shanghai, China). All cell lines were grown in Dulbecco's modified Eagle's medium (DMEM) supplementing with 10% FBS (ThermoFisher Scientific, Waltham, US) and penicillin/streptomycin in an incubator at 37°C and 5% $CO_2$.

## Establishment of stable JMJD2D-knockdown cell lines

Two different JMJD2D shRNA plasmids were purchased from Sigma-Aldrich (St. Louis. US). Control and JMJD2D shRNA plasmids were transfected into CMT93 cells using Lipofectamine 2000 (ThermoFisher Scientific, Waltham, US) according to the manufacturer's instructions. Puromycin (1 μg/mL) was used to select stable JMJD2D-knockdown cell lines.

## Western blot

Cultured control and JMJD2D-knockdown cells were lysed in RIPA buffer containing 1× Protease Inhibitor Cocktail (ThermoFisher Scientific, Waltham, US). Lysates were then centrifuged for 15 min at 13,000 rpm at 4°C and the supernatants were collected. Protein concentration was measured using BCA Assay. 20–40 μg total protein was loaded per lane, and isolated in Tris/glycine/SDS running buffer. Then, the proteins were transferred into PVDF membranes (Bio-Rad, Hercules, US). Membranes were blocked in 5% non-fat milk for 1 h and were incubated with the following primary antibodies shaking at 4°C overnight in blocking buffer: anti-JMJD2D (Abcam, Cat#ab93694), anti-STAT3 (#9139S; CST), anti-phosphorylation-STAT3 (#9145; CST), anti-p65 (#8242, CST), anti- phosphorylation-p65 (#3031, CST) and anti-β-actin (Sigma-Aldrich, Cat #A5441). Membranes were washed three times with TBST and incubated with horseradish peroxidase-conjugated secondary antibody (ThermoFisher Scientific, Waltham, US) and visualized by chemiluminescence.

## Quantitative real-time PCR (qRT-PCR) reaction

Total RNA was isolated from cells and colons using TRIZOL (ThermoFisher Scientific, Waltham, US) according to manufacturer's instructions. Complementary DNA was reversed using the ReverTra Ace qPCR RT Master Mix Kit (TOYOBO, Osaka, Japan) according to manufacturer's instructions. qRT-PCR was performed in a CFX96 apparatus (Bio-Rad, Hercules, CA) by using SYBR Green Master Mix (Applied Biosystems, Darmstadt, Germany). The relative levels of genes expression were normalized to housekeeping gene β-actin and were calculated with the ΔΔCt method. The RT-qPCR primers were listed as follow: Defensin-β1 F, 5′- AGGT GTTGGCATTCTCACAAG-3′, and Defensin-β1 R, 5′- GCTTATCTGGTTTACAGGTTCCC-3′; Defensin-β2 F, 5′- TATGCTGCCTCCTTTTCTCA-3′, and Defensin-β2 R, 5′- GACTTCCA TGTGCTTCCTTC-3′; Defensin-β3 F, 5′- GTCTCCACCTGCAGCTTTTAG-3′, and Defensin-β3 R, 5′- GTCTCCACCTGCAGCTTTTAG-3′; Defensin-β4 F, 5′- GCAGCCTTTACC CAAATTATC-3′, and Defensin-β4 R, 5′- ACAATTGCCAATCTGTCGAA-3′; IL-17F F, 5'-TGCTACTGTTGATGTTGGGAC-3', and IL-17F R, 5'-AATGCCCTGGTTTTGGTTGAA-3'; IL-17A F, 5'-GCTCCAGAAGGCCCTCAGA-3', and IL-17A R, 5'-CTTTCCCTCCGCA

TTGACA-3'; IL-17RA F, 5'- AGTGTTTCCTCTACCCAGCAC-3′, and IL-17RA R, 5′- GAAAACCGCCACCGCTTAC-3'; IL-17RC F, 5'- GCTGCCTGATGGTGACAATGT-3′, and IL-17RC R, 5'- GTCTGGCTCTAGCGACCAC-3'; Reg3β F, 5'- ATGGCTCCTACTGCT ATGCC-3', and Reg3β R, 5'-GTGTCCTCCAGGCCTCTTT-3'; Reg3γ F, 5'-TCAGGTGCAAG GTGAAGTTG-3', and R, 5'- GGCCACTGTTACCACTGCTT-3'; CRAMP F, 5'- CTTCAA CCAGCAGTCCCTAGACA-3', and CRAMP R, 5'- TCCAGGTCCAGGAGACGGTA-3'; RELM-β F, 5'- TGGCTTTGCCTGTGGATCTT-3', and RELM-β R, 5'- GCAGTGGTCCAGT CAACGAGTA-3'; S100A8 F, 5'- TGTCCTCAGTTTGTGCAGAATATAAA-3', and S100A8 R, 5'- TCACCATCGCAAGGAACTCC-3'; S100A9 F, 5'- AAAGGCTGTGGGAAGTAATTA AGAG-3', and S100A9 R, 5'- GCCATTGAGTAAGCCATTCCC-3'; Muc5Ac F, 5'- CCATG CAGAGTCCTCAGAACA-3', and Muc5Ac R, 5'-TTACTGGAAAGGCCCAAGCA-3'; IL-22 F, 5′-TCCGAGGAGTCAGTGCTAAA-3′, and IL-22 R, 5′-AGAACGTCTTCCAGGGTGAA-3'; IL-23 F, 5'- AGCGGGACATATGAATCTACTAAGAGA-3', and IL-23 R, 5'- GTCCTAG TAGGGAGGTGTGAAGTTG-3'; β-actin F, 5′-ACTATTGGCAACGAGCGGTTCC-3′, and β-actin R, 5′-GGCATAGAGGTCTTTACGGATGTCA-3′.

## Luciferase reporter assay

The IL-17F promoter reporter was transfected into CMT93 cells using Lipofectamine 2000 (ThermoFisher Scientific, Waltham, US) according to the manufacturer's instructions. The activities of IL-17F promoter reporter were analyzed by the luciferase reporter assay system (Promega, Madison, US) according to the manufacturer's instructions. The renilla reniformis luciferase reporter acted as an internal control.

## Co-IP assays

The interaction between JMJD2D and STAT3 in CMT93 cells was analyzed by the Co-IP assay as described previously [20]. The supernatants were immunoprecipitated with 2 μg of the following antibodies or control immunoglobulin G (IgG) with rotation for overnight at 4°C: anti-JMJD2D (Abcam, Cat#ab93694) and anti-STAT3 (#9139S; CST), and the immunoprecipitants were then used to western blot.

## ChIP assay

Control and JMJD2D-knockdown of CMT93 cells were treated with *C. rodentium* for 1 h or treated with mouse IL-22 for 24 hr, and then the chromatin was immunoprecipitated using anti-JMJD2D, anti-STAT3, H3K9me3 or nonspecific IgG (Santa Cruz, Dallas, US) antibody. ChIP DNAs were purified and amplified by qRT-PCR with specific primers for the IL-17F and Reg3β promoter. The IL-17F promoter primers were as follow: F, 5'-GAGAGTGTTTTCAAA AGCTGAGTAA-3'; R, 5'-AAAATCAGAATGAAGCATGAAATAA-3'. The Reg3β promoter primers were as follow: F, 5'- GGAGAGGAACCCATCTACTGC-3'; R, 5'- ATAGGGCAAC TTCACCTCACA-3'.

## Statistical analysis

All bar charts and statistical analyses were performed using GraphPad Prism 5. All data were shown as the Mean + SEM. The statistically significant effects between mean values ($P < 0.05$) were assessed with the two-tailed Student's t-test.

## Results

### JMJD2D<sup>-/-</sup> mice are more sensitive to *C. rodentium* infection compared to wild-type mice

To define the role of JMJD2D in *C. rodentium*-induced colitis, we first detected the expression of JMJD2D in the colonic epithelial cells of wild-type mice orally infected with *C. rodentium*. As shown in **Fig 1A**, JMJD2D expression was markedly increased in the colonic epithelial cells isolated from mice on days 7 and 14 after *C. rodentium* infection, implying that JMJD2D in the colonic epithelial cells may be participated in the clearance of *C. rodentium*. We then orally infected wild-type and JMJD2D<sup>-/-</sup> mice with *C. rodentium* and measured body weight and bacterial burdens in the feces, colons, spleens, lungs, and livers for up to 28 days. JMJD2D<sup>-/-</sup> mice exhibited more body weight loss on days 10–14 after *C. rodentium* infection compared to wild-type mice (**Fig 1B**). Furthermore, JMJD2D<sup>-/-</sup> mice displayed 10–100 folds higher *C. rodentium* burdens in the feces, colons, and cecums + feces compared to wild-type mice on days 7, 14, and 21 after infection (**Fig 1C**). The *C. rodentium* burdens in the spleens, lungs, and livers of JMJD2D<sup>-/-</sup> mice were also increased compared to wild-type mice on days 7, 14, and 21 after infection, suggesting that JMJD2D<sup>-/-</sup> mice exhibit more severe systemic dissemination of *C. rodentium* compared to wild-type mice. Consistently, immunohistochemical staining of *C. rodentium* showed that more *C. rodentium* bacteria, which penetrated deeply into crypts, were observed in the colons of JMJD2D<sup>-/-</sup> mice compared to wild-type mice on days 7 and 14 after infection (**Fig 1D**). These results indicated that JMJD2D plays an important protective role in host defense against *C. rodentium* infection.

*C. rodentium* infection leads to colonic inflammation and injury [21]. A shorter colon length, which is a feature of colonic inflammation and injury, was observed in JMJD2D<sup>-/-</sup> mice compared to wild-type mice on days 7 and 14 after infection (**Fig 1E**). H&E staining was performed to score colonic pathology in the colons of JMJD2D<sup>-/-</sup> mice and wild-type mice without or with *C. rodentium* infection. Histological analysis of colons from JMJD2D<sup>-/-</sup> mice infected with *C. rodentium* showed more serious submucosal inflammation and edema compared to wild-type mice, resulting in significantly higher histological scores in the colons (**Fig 1F**). Furthermore, the levels of TNF-α and IL-6 in the colonic culture supernatant of JMJD2D<sup>-/-</sup> mice on day 14 after infection were significantly more than those in the colonic culture supernatants of wild-type mice (**Fig 1G**). Taken together, JMJD2D<sup>-/-</sup> mice exhibited more serious colonic pathology and produced more inflammatory cytokines compared to wild-type mice after *C. rodentium* infection.

### JMJD2D expression in non-bone marrow-derived cells controls *C. rodentium* infection

To determine which JMJD2D-expressing cells are important for the control of *C. rodentium* infection, we performed reciprocal bone marrow transfer experiments between JMJD2D<sup>-/-</sup> mice and wild-type mice. Mice were orally infected with *C. rodentium* 8 weeks after bone marrow transfer and monitored the body weight and *C. rodentium* burdens in the feces, colons, and spleens for 7 days. The *C. rodentium* burdens in the feces from WT BM → WT mice were lower than those from WT BM → JMJD2D<sup>-/-</sup> mice, but were comparable with those from JMJD2D<sup>-/-</sup> BM→ WT mice on days 7 after infection (**Fig 2A**). Additionally, the *C. rodentium* burdens in the colons from JMJD2D<sup>-/-</sup> BM → JMJD2D<sup>-/-</sup> mice were higher than those from JMJD2D<sup>-/-</sup> BM → WT mice, but were comparable with those from WT BM → JMJD2D<sup>-/-</sup> mice on days 7 after infection (**Fig 2B**). Similar results were also observed in the spleens (**Fig 2C**). Consistently, anti-*C. rodentium* staining showed that more *C. rodentium* bacteria were

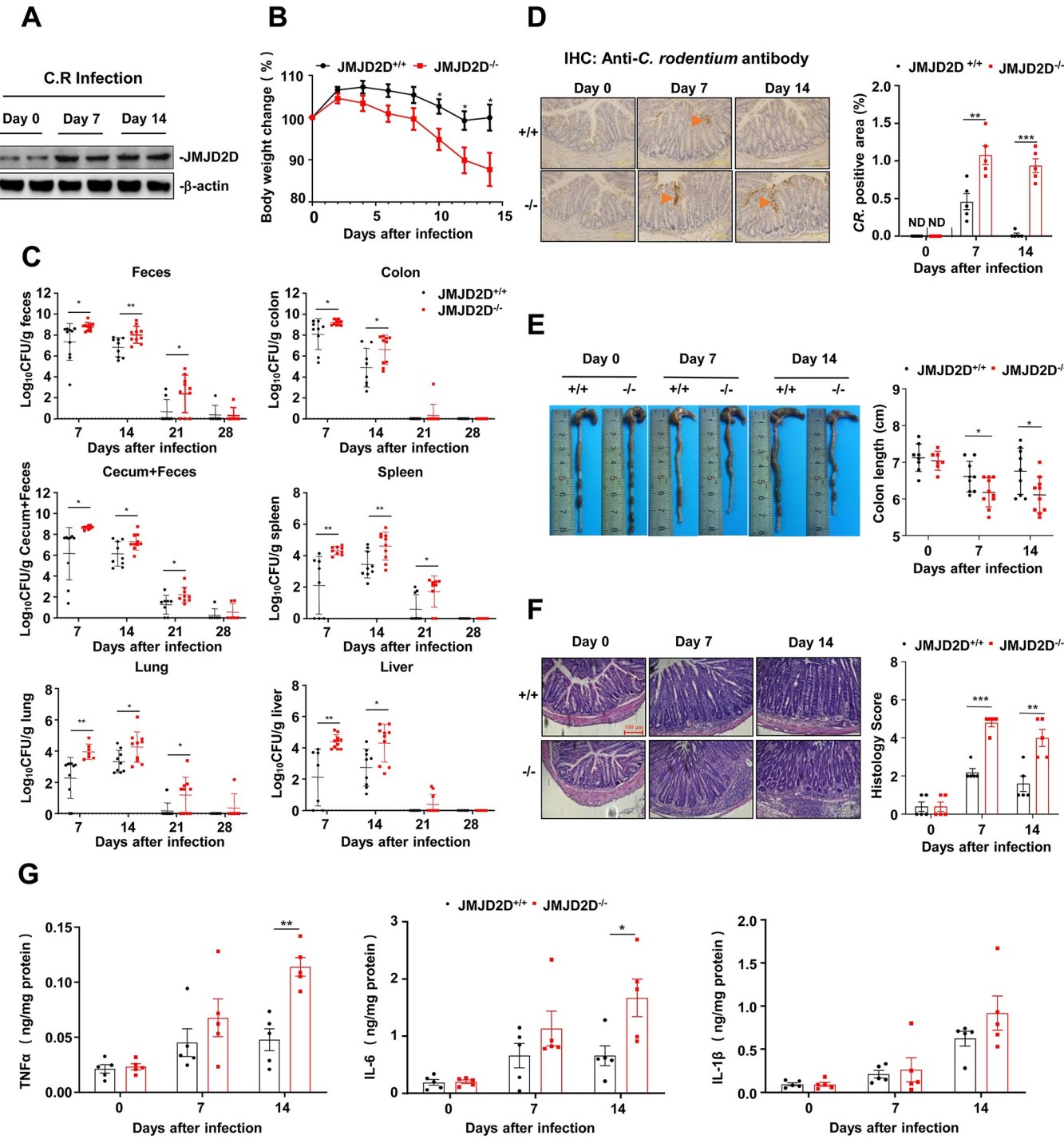

**Fig 1. JMJD2D⁻ᐟ⁻ mice are more sensitive to *C. rodentium* infection compared to wild-type mice.** (**A**) JMJD2D expression in mouse colons was increased after *C. rodentium* infection. The levels of JMJD2D in colonic epithelial cells after oral administration of *C. rodentium* were assessed by western blot analysis. (**B**) JMJD2D⁻ᐟ⁻ mice (n = 9) exhibited more body weight loss on days 10–14 compared to wild-type mice (n = 9) after *C. rodentium* infection. (**C**) JMJD2D⁻ᐟ⁻ mice displayed more *C. rodentium* burdens in the feces, colons, cecums + feces, spleens, lungs, and livers compared to wild-type mice on days 7, 14, and 21 after *C. rodentium* infection. (**D**) More *C. rodentium* bacterias were observed in the colons of JMJD2D⁻ᐟ⁻ mice compared to wild-type mice on days 7 and 14 after infection. Representative anti-*C. rodentium* IHC staining of the colon sections of JMJD2D⁻ᐟ⁻ mice (n = 5) and wild-type (n = 5) mice on days 0, 7, 14 after *C. rodentium* infection. Arrow and arrowhead denote *C. rodentium*. Pictures are representative of three independent experiments. (**E**) A shorter colon length was observed in JMJD2D⁻ᐟ⁻ mice compared to wild-type mice on days 7 and 14 after *C. rodentium* infection. Colon length of JMJD2D⁻ᐟ⁻ mice and wild-type mice (n = 5–10) on days 7, 14, 21 and 28 days after *C. rodentium* infection. (**F**) JMJD2D⁻ᐟ⁻ mice showed more serious submucosal inflammation and edema compared to wild-type mice after *C. rodentium* infection. H&E staining of colon sections from JMJD2D⁻ᐟ⁻ mice and wild-type mice (n = 5) on days 7, 14 and 21

days after *C. rodentium* infection. (**G**) The levels of TNF-α and IL-6 in the colonic culture supernatant of JMJD2D⁻/⁻ mice on day 14 after infection were significantly more than those in the colonic culture supernatants of wild-type mice. The protein levels of TNFα, IL-6 and IL-1β in the colons from JMJD2D⁻/⁻ mice and wild-type mice (n = 5) on days 7, 14 and 21 days after *C. rodentium* infection. Data are shown as mean±SEM. Results are representative of three independent experiments. *P<0.05; **P<0.01; ***P<0.001.

observed in the colons from JMJD2D⁻/⁻ BM → JMJD2D⁻/⁻ mice compared to those from JMJD2D⁻/⁻ BM → WT mice, but were comparable with those from WT BM → JMJD2D⁻/⁻ mice on days 7 after infection (**Fig 2D**). Furthermore, histological analysis of colons from WT BM → JMJD2D⁻/⁻ mice and JMJD2D⁻/⁻ BM → JMJD2D⁻/⁻ mice infected with *C. rodentium* showed higher histology score compared to WT BM → WT mice and JMJD2D⁻/⁻ BM→ WT mice (**Fig 2E**). Taken together, these results indicated that JMJD2D expression in non-bone marrow-derived cells plays a critical role in controlling *C. rodentium* infection.

## JMJD2D deficiency impairs the expression of β-defensins and IL-17F in the colonic epithelial cells during *C. rodentium* infection

Due to JMJD2D expression in non-bone marrow-derived cells controls *C. rodentium* infection, we hypothesized that JMJD2D expression in the colonic epithelial cells plays a critical role in controlling *C. rodentium* infection. To test it, we examined the expression of antimicrobial genes such as Reg family and β-defensins, which play critical roles in controlling *C. rodentium* infection [7,8], in the colonic epithelial cells of wild-type and JMJD2D⁻/⁻ mice infected with *C. rodentium*. As shown in **S2 Fig**, there was no significant difference in the expression of Reg 3β, Reg 3γ, CRAMP, RELM-β, S100A8, S100A9, and Muc5Ac in the colonic epithelial cells of wild-type and JMJD2D⁻/⁻ mice infected with *C. rodentium*; however, JMJD2D deficiency impaired the expression of β-defensin-1, β-defensin-3, and β-defensin-4, but not β-defensin-2, in the colonic epithelial cells of mice without or with *C. rodentium* infection (**Fig 3A–3D**). These results indicated that JMJD2D expression in the colonic epithelial cells promotes the expression of β-defensin-1, β-defensin-3, and β-defensin-4 to control *C. rodentium* infection.

It has been reported that IL-17A and IL-17F can induce the expression of β-defensins [8]. Therefore we detected the expression of IL-17A and IL-17F in the colonic epithelial cells of wild-type and JMJD2D⁻/⁻ mice infected with *C. rodentium*. The expression of IL-17F in the colonic epithelial cells of JMJD2D⁻/⁻ mice was significantly reduced compared to wild-type mice on days 0, 7 and 14 after *C. rodentium* infection (**Fig 3E**), whereas the expression of IL-17A in the epithelial cells was comparable between wild-type and JMJD2D⁻/⁻ mice (**Fig 3F**). Consistent with the mRNA levels of IL-17F, the protein levels of IL-17F in the colonic culture supernatants of JMJD2D⁻/⁻ mice were markedly decreased compared to those of wild-type mice on days 0, 7, and 14 after *C. rodentium* infection (**Fig 3G**). JMJD2D deficiency did not impair the expression of IL-17 receptors IL-17RA and IL-17RC in the colonic epithelial cells (**Fig 3H and 3I**). Taken together, these results implicated that JMJD2D expression in the colonic epithelial cells may promote the expression of β-defensin-1, β-defensin-3, and β-defensin-4 during *C. rodentium* infection by enhancing IL-17F expression.

## JMJD2D promotes the expression of β-defensins by enhancing IL-17F expression in colorectal cancer cells

To verify that IL-17F can promote β-defensin expression, we transfected IL-17F specific small interfering RNA to knock down IL-17F expression in murine colorectal cancer cell line CMT93 and then examined the effects of IL-17F knockdown on β-defensin expression. As shown in **Fig 4A**, the expression of β-defensin-1, β-defensin-3, and β-defensin-4 was markedly decreased in IL-17F-knockdown CMT93 cells, demonstrating that IL-17F can promote β-

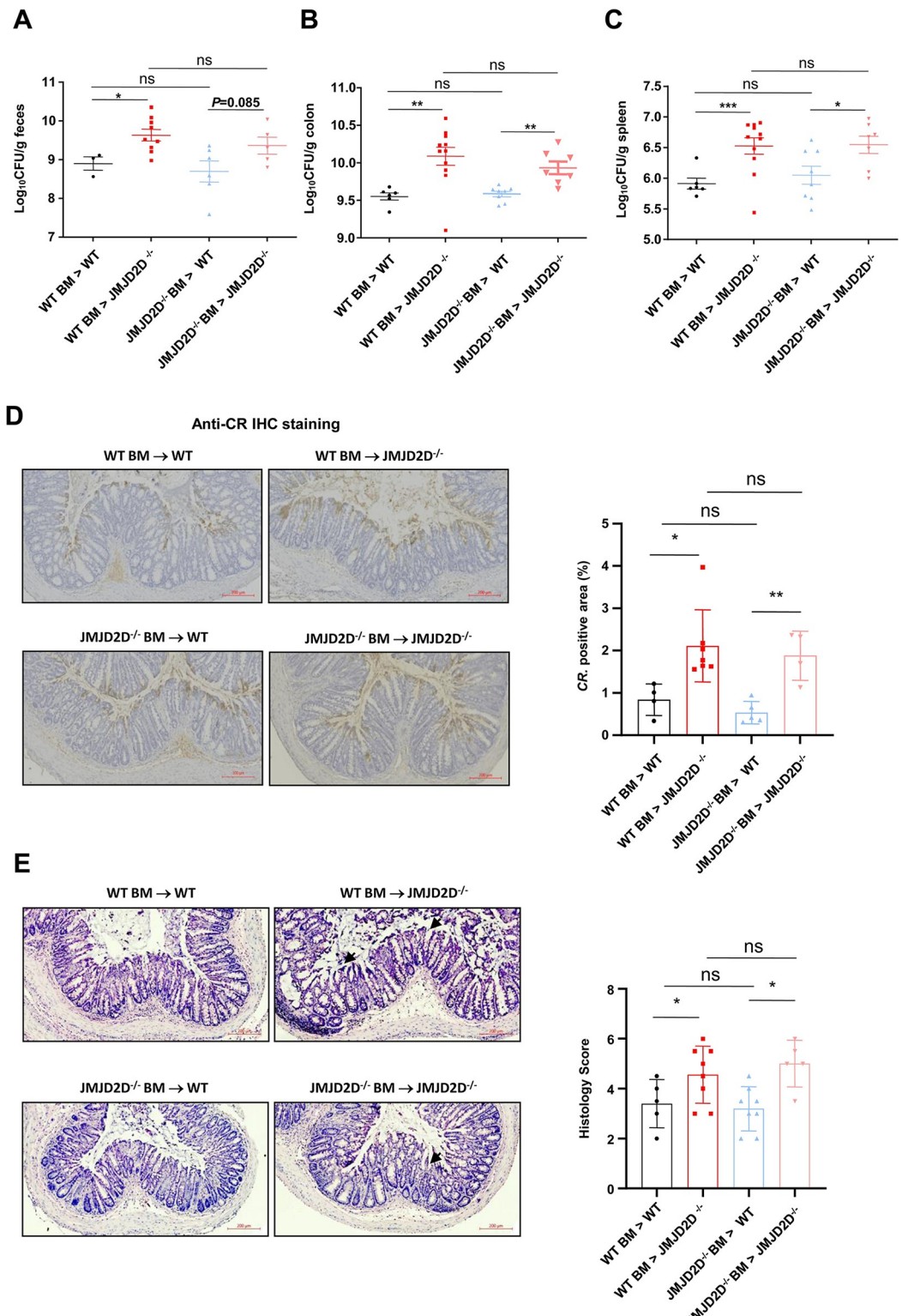

**Fig 2. JMJD2D expression in non-bone marrow-derived cells controls *C. rodentium* infection.** (**A-C**) The *C. rodentium* burdens in the feces (**A**), colons (**B**), and spleens (**C**) from WT BM → WT mice were lower than those from WT BM → JMJD2D$^{-/-}$ mice, but were comparable with those from JMJD2D$^{-/-}$ BM→ WT mice on days 7 after infection. (**D**) More *C. rodentium* bacteria were observed in the colons from JMJD2D$^{-/-}$ BM → JMJD2D$^{-/-}$ mice compared to those from JMJD2D$^{-/-}$ BM → WT mice, but were comparable with those from WT BM → JMJD2D$^{-/-}$ mice on days 7 after infection.

n = 4–7. Brown signals represent *C. rodentium*. Representative pictures of anti-*C. rodentium* immunohistochemistry staining of the colon sections (Left panel); Quantification of *C. rodentium*-positive area in chimeric mice (Right panel). Scale bars represent 200 μm. (**E**) Histological analysis of colons from WT BM → JMJD2D⁻/⁻ mice and JMJD2D⁻/⁻ BM → JMJD2D⁻/⁻ mice infected with *C. rodentium* showed higher histology score compared to WT BM → WT mice and JMJD2D⁻/⁻ BM→ WT mice. n = 5–8. Representative pictures of H&E staining of the colon sections (Left panel); Quantification of histopathology scores (Right panel). Scale bars represent 200 μm. Data are shown as mean±SEM. n.s. represents no significant difference. *$P<0.05$; **$P<0.01$; ***$P<0.001$.

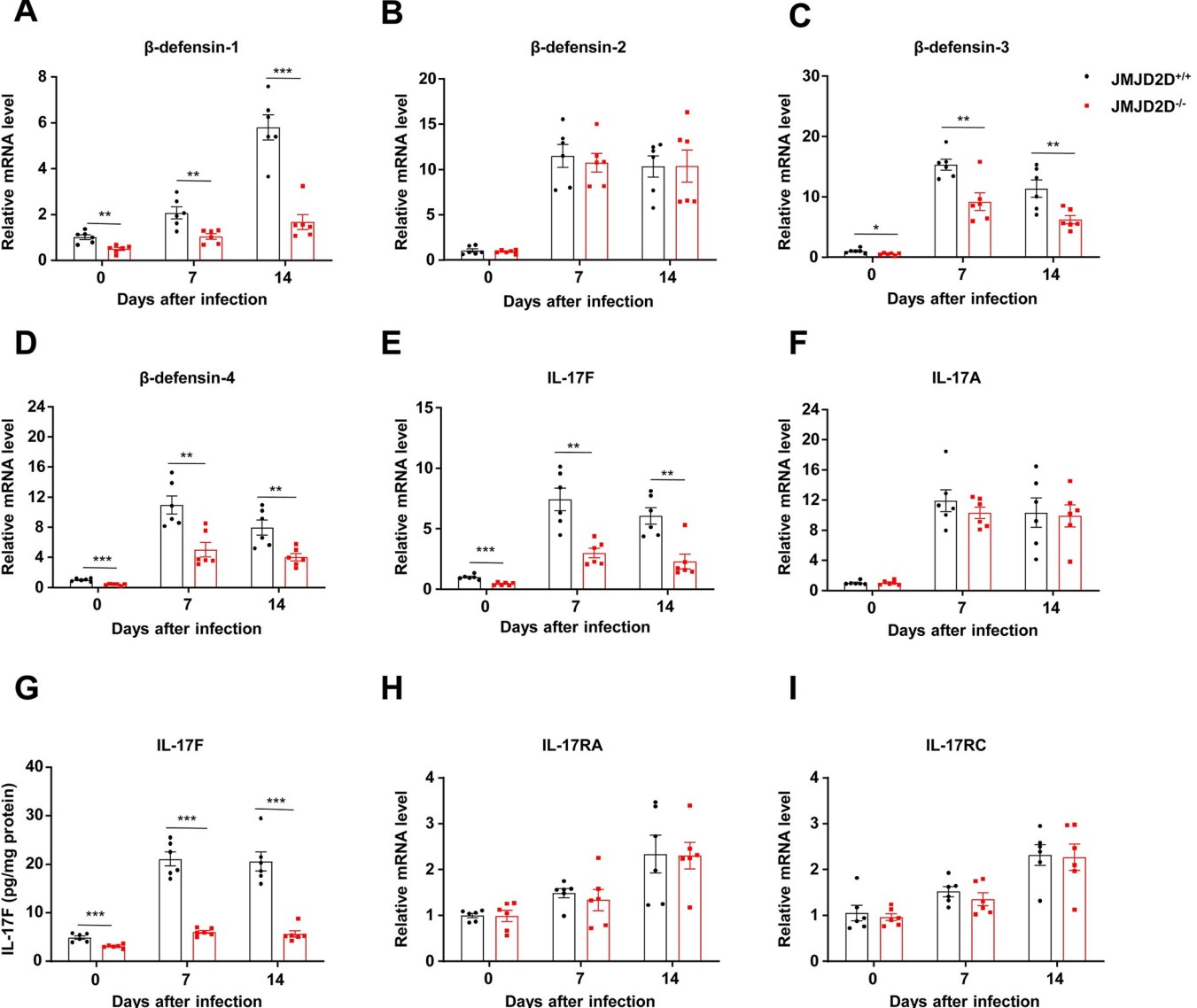

**Fig 3. JMJD2D deficiency impairs the expression of β-defensins and IL-17F in the colonic epithelial cells during *C. rodentium* infection.** (**A-D**) JMJD2D deficiency impaired the expression of β-defensin-1, β-defensin-3, and β-defensin-4, but not β-defensin-2, in the colonic epithelial cells of mice without or with *C. rodentium* infection. (**E-F**) The expression of IL-17F in the colonic epithelial cells of JMJD2D⁻/⁻ mice was significantly reduced compared to wild-type mice on days 0, 7 and 14 after *C. rodentium* infection (**E**), whereas the expression of IL-17A in the epithelial cells was comparable between wild-type and JMJD2D⁻/⁻ mice (**F**). (**G**) The protein levels of IL-17F in the colonic cultured supernatant of JMJD2D⁻/⁻ mice were markedly decreased compared to those of wild-type mice on days 0, 7 and 14 after *C. rodentium* infection. (**H-I**) JMJD2D deficiency did not impair the expression of IL-17 receptors IL-17RA and IL-17RC in the colonic epithelial cells. n = 6. Data are shown as mean±SEM. Results are representative of three independent experiments. **$P<0.01$; **$P<0.001$.

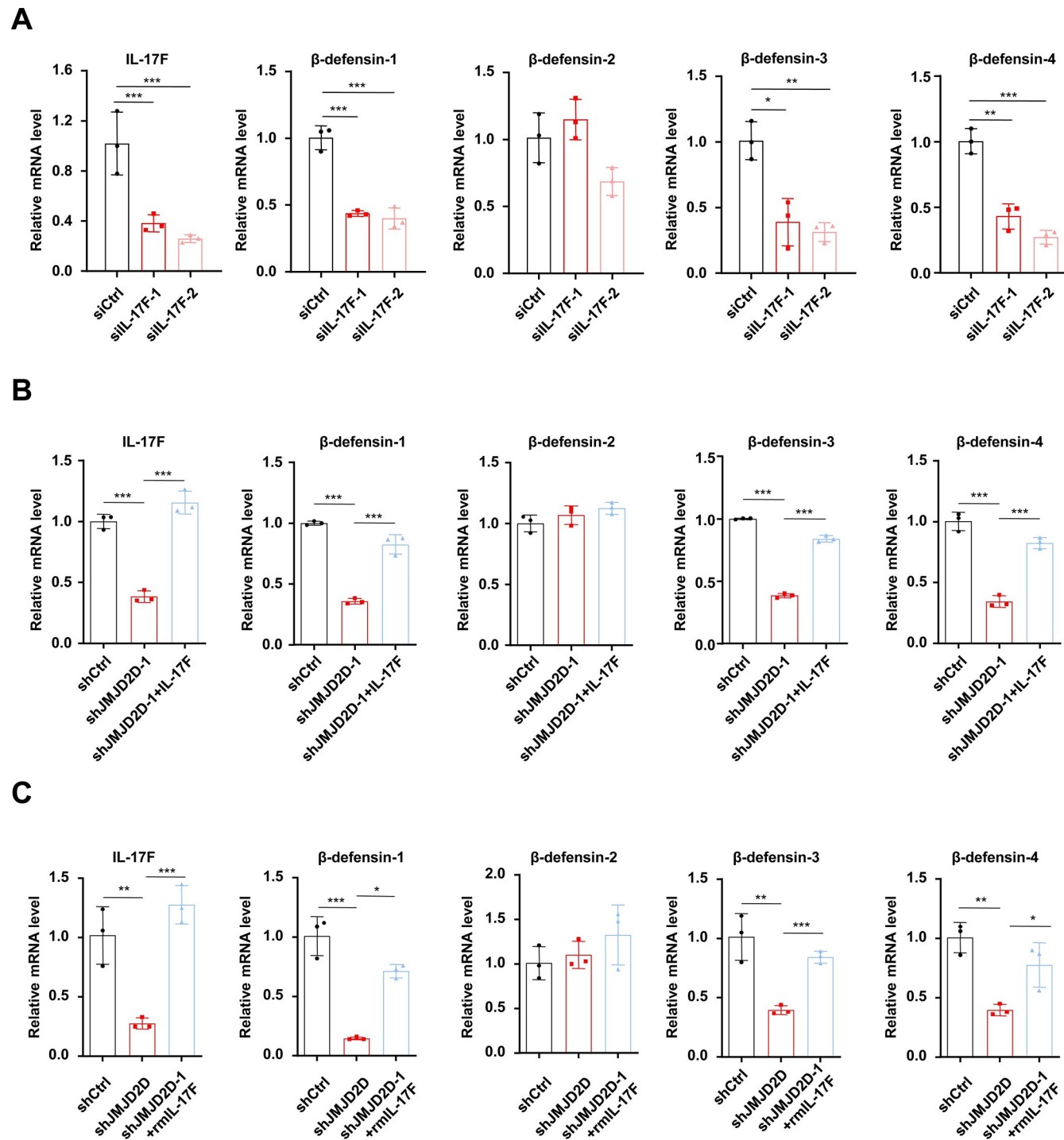

**Fig 4. JMJD2D promotes the expression of β-defensins by enhancing IL-17F expression in colon carcinoma cells.** (**A**) The expression of β-defensin-1, β-defensin-3, and β-defensin-4 was markedly decreased in IL-17F-knockdown CMT93 cells. (**B**) The expression of IL-17F, β-defensin-1, β-defensin-3, and β-defensin-4 in JMJD2D-knockdown CMT93 cells was reduced, but ectopic IL-17F expression in JMJD2D–knockdown CMT93 cells rescued the expression of IL-17, β-defensin-1, β-defensin-3, and β-defensin-4. (**C**) Recombinant IL-17F treatment rescued the expression of IL-17F, β-defensin-1, β-defensin-3, and β-defensin-4 in JMJD2D-knockdown CMT93 cells. Data are shown as mean±SD. The experiments were repeated three times independently. *$P<0.05$; **$P<0.01$; ***$P<0.001$.

defensin expression in colorectal cancer cells. Similar to the *in vivo* results, the expression of IL-17F, β-defensin-1, β-defensin-3, and β-defensin-4 in JMJD2D-knockdown CMT93 cells was reduced (**Fig 4B**). To confirm that JMJD2D-promoted expression of β-defensins was mediated by IL-17F, we rescued IL-17F expression in JMJD2D-knockdown CMT93 cells. As shown in **Fig 4B**, ectopic IL-17F expression in JMJD2D–knockdown CMT93 cells rescued the expression of β-defensin-1, β-defensin-3, and β-defensin-4. Furthermore, we treated JMJD2D-knockdown CMT93 cells with recombinant IL-17F. Similar to the results of ectopic IL-17F expression, recombinant IL-17F treatment rescued the expression of β-defensin-1, β-defensin-3, and β-defensin-4 in JMJD2D-knockdown CMT93 cells (**Fig 4C**). Taken together, these results demonstrated that JMJD2D promotes the expression of β-defensins by enhancing IL-17F expression.

## JMJD2D promotes IL-17F expression via enhancing STAT3 signaling

To reveal the mechanisms by which JMJD2D promotes IL-17F expression to further induce β-defensin expression in response to *C. rodentium* infection, we treated wild-type and JMJD2D-knockdown CMT93 cells with heat-killed *C. rodentium* and then examined the expression of JMJD2D, IL-17F, and β-defensins. We first examined the effect of heat-killed *C. rodentium* treatment on the expression of JMJD2D. As shown in **Fig 5A**, heat-killed *C. rodentium* treatment induced JMJD2D expression. We previously showed that JMJD2D expression could be induced by activating NF-κB signaling [17], we therefore hypothesized that heat-killed *C. rodentium* treatment may activate NF-κB signaling to induce JMJD2D expression. Indeed, heat-killed *C. rodentium* treatment could activate NF-κB signaling as demonstrated by increased phosphorylation of p65 (**S3A Fig**), suggesting that JMJD2D expression is induced by activated NF-κB signaling in response to *C. rodentium* infection. We next examined the effect of heat-killed *C. rodentium* treatment on the expression of IL-17F and β-defensins. As shown in **Fig 5B**, heat-killed *C. rodentium* treatment markedly induced the mRNA expression of IL-17F, β-defensin-1, β-defensin-3, and β-defensin-4, but the induction was significantly reduced in JMJD2D-knockdown CMT93 cells. Furthermore, heat-killed *C. rodentium* induced IL-17F promoter reporter activity about 7-fold in wild-type CMT93 cells, whereas the induction of IL-17F promoter reporter activity was dramatically decreased to 2-fold in JMJD2D-knockdown CMT93 cells (**Fig 5C**). These results suggested that JMJD2D promotes IL-17F expression at the transcriptional levels.

To find out which transcription factor JMJD2D can cooperate with to promote IL-17F expression, we screened the sequence of mouse IL-17F promoter in Jaspar databases and PROMO databases to search for the potential transcription factor binding sites. We found a transcription factor STAT3 binding site (GTTCTAAGAAA, -1825~-1815) on the IL-17F promoter, implicating that STAT3 may bind to this site to induce IL-17F expression. Additionally, western blotting results showed that heat-killed *C. rodentium* treatment activated STAT3 signaling as demonstrated by increased phosphorylation of STAT3 in control CMT93 cells, whereas the induction of STAT3 phosphorylation was reduced in JMJD2D-knockdown CMT93 cells (**Fig 5D**). Similarly, STAT3 phosphorylation was impaired in the colonic epithelial cells of JMJD2D$^{-/-}$ mice on days 7 and 14 after *C. rodentium* infection (S3B Fig). Furthermore, Co-IP assays showed that JMJD2D could interact with STAT3 (**Fig 5E**), and the interaction was enhanced by heat-killed *C. rodentium* treatment (**Fig 5E**). In light of these results, we hypothesized that JMJD2D may cooperate with STAT3 to induce IL-17F expression. To test this hypothesis, we co-transfected CMT93 cells with IL-17F promoter reporter and/or the expression vectors of JMJD2D, STAT3, and JMJD2D plus STAT3, respectively. Transfection of STAT3 induced IL-17F promoter reporter activity 2.7-fold (**Fig 5F**), suggesting

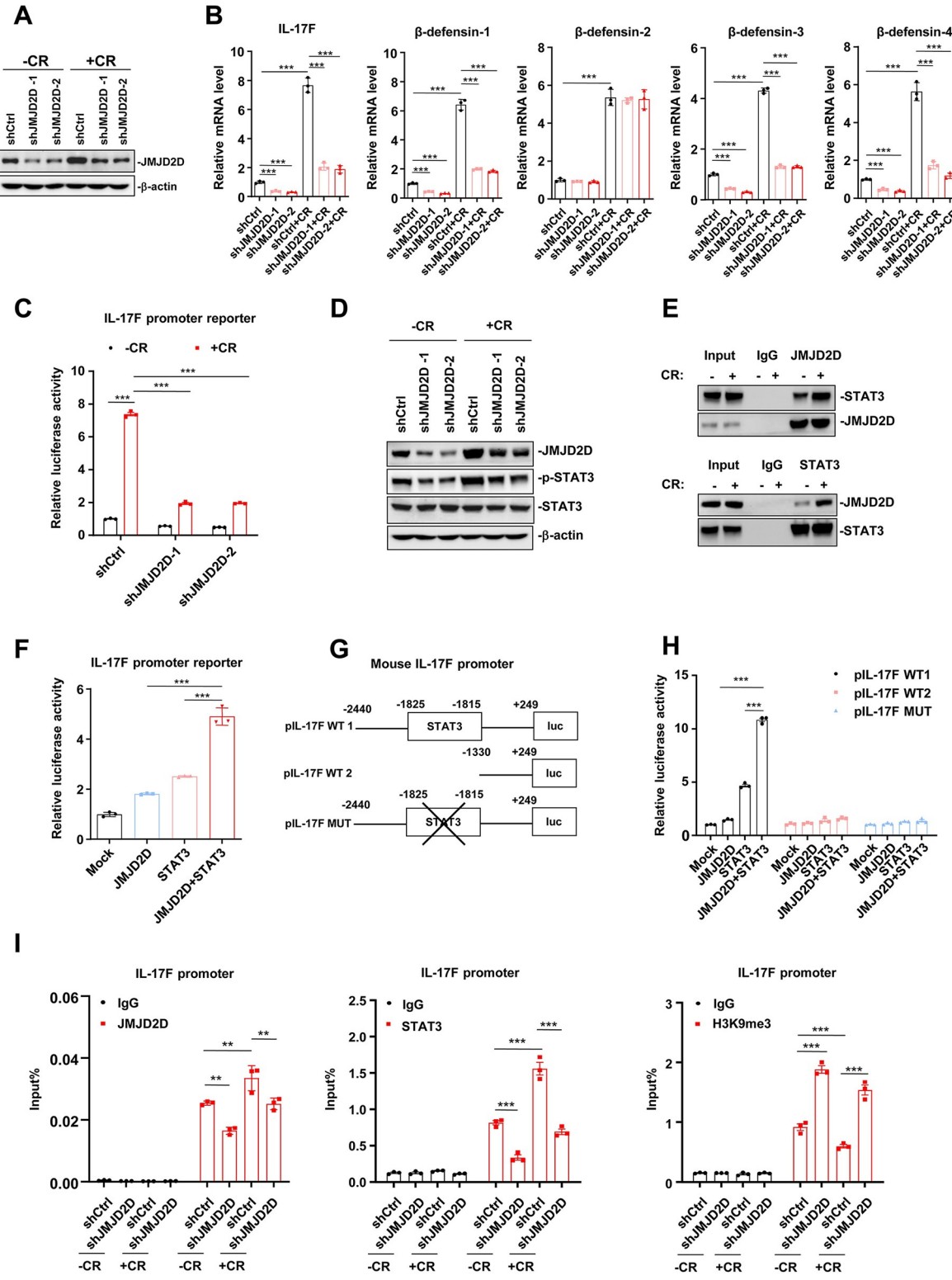

**Fig 5. JMJD2D promotes IL-17F expression via enhancing STAT3 signaling.** (**A**) Heat-killed *C. rodentium* treatment induced JMJD2D expression. (**B**) Heat-killed *C. rodentium* treatment markedly induced the mRNA expression of IL-17F, β-defensin-1, β-defensin-3, and β-defensin-4, but the induction was significantly reduced in JMJD2D-knockdown CMT93 cells. (**C**) Heat-killed *C. rodentium* (MOI = 200) for 2 h induced IL-17F promoter reporter activity about 7-fold in wild-type CMT93 cells, whereas the induction of IL-17F promoter reporter activity was dramatically decreased to 2-fold in JMJD2D-knockdown CMT93 cells. (**D**) Heat-killed *C. rodentium* treatment

(MOI = 200) for 2 h activated STAT3 signaling as demonstrated by increased phosphorylation of STAT3 in control CMT93 cells, whereas the induction of STAT3 phosphorylation was reduced in JMJD2D-knockdown CMT93 cells. (**E**) JMJD2D could interact with STAT3 and the interaction was enhanced by heat-killed *C. rodentium* treatment. Co-IP analysis of the interaction of JMJD2D and STAT3 in CMT93 cells. (**F**) Co-transfection of JMJD2D and STAT3 synergistically induced IL-17F promoter reporter activity. (**G**) Schematic maps of different IL-17F promoter constructs. (**H**) Co-transfection of JMJD2D and STAT3 synergistically induced the activity of wild-type IL-17F promoter reporter (with STAT3 binding site), whereas it failed to induce the activity of STAT3 binding site-absented or STAT3 binding site-mutated IL-17F promoter reporter. (**I**) Knockdown of JMJD2D reduced the levels of STAT3 and JMJD2D on IL-17F promoters in CMT93 cells with or without heat-killed *C. rodentium* treatment (MOI = 200) for 2 h, but increased H3K9me3 levels on IL-17F promoter in CMT93 cells with or without heat-killed *C. rodentium* treatment. Data are shown as mean±SEM. Results are representative of three independent experiments. **$P<0.01$; ***$P<0.001$.

that STAT3 can activate IL-17F transcription. Transfection of JMJD2D only induced IL-17F promoter reporter activity 1.8-fold, while co-transfection of JMJD2D and STAT3 synergistically induced IL-17F promoter reporter activity 5-fold (**Fig 5F**), indicating that JMJD2D can cooperate with STAT3 to enhance IL-17F expression.

To determine the role of STAT3 binding site in STAT3-induced IL-17F expression, we performed promoter reporter mutation assays. Co-transfection of JMJD2D and STAT3 could synergistically induce the activity of wild-type IL-17F promoter reporter (with STAT3 binding site), whereas it failed to induce the activity of STAT3 binding site-absent or STAT3 binding site-mutated IL-17F promoter reporter (**Fig 5G** and **5H**). To define whether JMJD2D and STAT3 could be recruited to the STAT3 binding site on the IL-17F promoter without or with heat-killed *C. rodentium* treatment, we performed ChIP assays. As shown in **Fig 5I**, STAT3 and JMJD2D could be recruited to the STAT3 binding site on the IL-17F promoter without heat-killed *C. rodentium* treatment, and the recruitment of JMJD2D and STAT3 on the IL-17F promoter was increased after heat-killed *C. rodentium* treatment. JMJD2D knockdown reduced the recruitment of JMJD2D and STAT3 on the IL-17F promoter without or with heat-killed *C. rodentium* treatment (**Fig 5I**). Additionally, JMJD2D knockdown dramatically elevated H3K9me3 levels on the IL-17F promoter (**Fig 5I**), suggesting that JMJD2D is responsible for demethylating H3K9me3 on the IL-17F promoter. Taken together, these results suggested that JMJD2D promotes IL-17F expression by enhancing STAT3 signaling via increasing STAT3 phosphorylation and recruitment on the STAT3 binding site on the IL-17F promoter to demethylate H3K9me3.

## The demethylase activity of JMJD2D is required for the induction of IL-17F expression

To determine whether the demethylase activity of JMJD2D is required for the induction of IL-17F transcription, we examined the effects of wild-type JMJD2D and demethylase-defective mutant JMJD2D-S200M on STAT3 phosphorylation and the expression of IL-17F and β-defensins in CMT93 cells. Overexpression of wild-type JMJD2D could enhance STAT3 phosphorylation, but overexpression of JMJD2D-S200M couldn't enhance STAT3 phosphorylation (**Fig 6A**). Overexpression of wild-type JMJD2D could increase the mRNA levels of IL-17F, β-defensin-1, β-defensin-3, and β-defensin-4, but overexpression of JMJD2D-S200M couldn't increase the expression of these genes (**Fig 6B**). These results suggested that the demethylase activity of JMJD2D is required for promoting the expression of IL-17F, β-defensin-1, β-defensin-3, and β-defensin-4 in CMT93 cells. We next examined the effects of wild-type JMJD2D and demethylase-defective mutant JMJD2D-S200M on IL-17F promoter reporter activity in the absence or presence of STAT3. As shown in **Fig 6C**, overexpression of JMJD2D increased IL-17 promoter reporter activity in the absence or presence of STAT3, but overexpression of JMJD2D-S200M couldn't increase IL-17 promoter reporter activity. Furthermore, overexpression of JMJD2D decreased H3K9me3 levels on the IL-17F promoter, but overexpression of

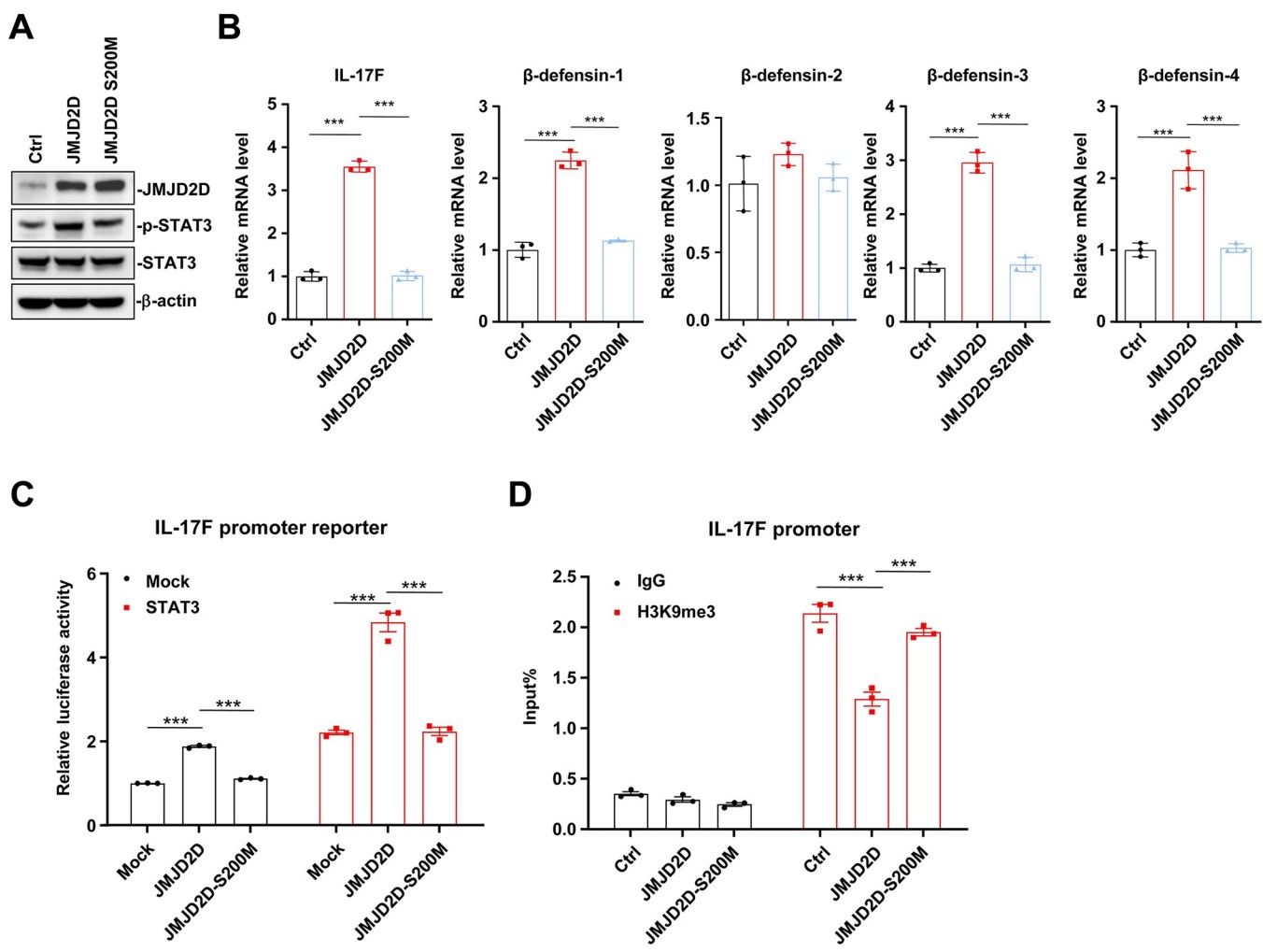

**Fig 6. The demethylase activity of JMJD2D is required for the up-regulation of IL-17F.** (**A**) Overexpression of wild-type JMJD2D could enhance STAT3 phosphorylation, but overexpression of JMJD2D-S200M could not. (**B**) Overexpression of wild-type JMJD2D could increase the mRNA levels of IL-17F, β-defensin-1, β-defensin-3, and β-defensin-4, but overexpression of JMJD2D-S200M could not. (**C**) Overexpression of JMJD2D increased IL-17 promoter reporter activity in the absence or presence of STAT3, but overexpression of JMJD2D-S200M could not. (**D**) Overexpression of JMJD2D decreased H3K9me3 levels on the IL-17F promoter, but overexpression of JMJD2D-S200M could not. Data are shown as mean±SEM. Results are representative of three independent experiments. ***$P<0.001$.

JMJD2D-S200M could not decrease H3K9me3 levels (**Fig 6D**). These results suggested that the demethylase activity of JMJD2D is required for the induction of IL-17F and β-defensins.

## Discussion

In this study, we disclosed that JMJD2D plays an important protective role in *C. rodentium*-induced colitis. Primarily, JMJD2D[-/-] mice displayed an impaired clearance of *C. rodentium* and more severe tissue pathology compared with wild-type mice. And JMJD2D[-/-] mice exhibited an impaired induction of IL-17F, β-defensin-1, β-defensin-3, and β-defensin-4 in the colonic epithelial cells during *C. rodentium* infection. At the molecular level, JMJD2D cooperates with STAT3 to enhance IL-17F expression to further induce the expression of β-defensin-1, β-defensin-3, and β-defensin-4.

It is known that hemopoietic cells including neutrophils, monocytes/macrophages, CD4[+] T cells, and dendritic cells play a critical role in clearing *C. rodentium* infection [22–25]. In addition, nonhemopoietic cells, predominantly the colonic epithelial cells, are also important in control of *C. rodentium* infection [26]. It has been reported that the mRNA levels of JMJD2D were highly expressed in hematopoietic system (bone marrow, lymph nodes, and spleen), hematopoietic cells (dendritic cells and macrophages) [27], and colonic epithelial cells [17]. In this study, to determine in which cells JMJD2D contributes to host defense against *C. rodentium* infection, we performed bone marrow transplantation experiments in wild-type and JMJD2D[-/-] mice. We found that JMJD2D[-/-] recipients that received bone marrow from wild-type or JMJD2D[-/-] mice exhibited more *C. rodentium* burdens in the colons and spleens compared to wild-type recipient mice that received bone marrow from wild-type or JMJD2D[-/-] mice on day 7 after *C. rodentium* infection, suggesting that JMJD2D in nonhemopoietic cells, most likely in colonic epithelial cells, is required for clearance of *C. rodentium*.

Innate immunity, including the intestinal barrier comprised of mucus layer, epithelial cells, and the nonepithelial cell (most notable leukocytes) [28], is the first line of defense against pathogens. Antimicrobial peptides secreted by intestinal epithelial cells, which can directly kill or inactivate bacteria, are key effectors of innate immunity [29,30]. Some antimicrobial peptides, including defensins, cathelicidins, and regenerating islet-derived protein (REG) family, have been found to contribute to host defense against *C. rodentium* infection [7,30,31]. Notably, β-defensins play a key role in host defense against *C. rodentium* infection [30,32]. In this study, we found that the *C. rodentium* burdens in JMJD2D[-/-] mice were higher than those in wild-type mice on days 7 and 14 after *C. rodentium* infection, and meanwhile the expression of β-defensin-1, β-defensin-3, and β-defensin-4 in the colonic epithelial cells of JMJD2D[-/-] mice was significantly decreased compared with wild-type mice. Thus, we propose that higher *C. rodentium* burdens in JMJD2D[-/-] mice are caused by a reduced innate immunity response, such as expression of β-defensin-1, β-defensin-3, and β-defensin-4 after *C. rodentium* infection, which lead to more severe colonic injury.

It has been reported that the expression of β-defensin-1, β-defensin-3, and β-defensin-4 is significantly decreased in the colons of IL-17F[-/-], IL-17A[-/-], and IL-17A[-/-];IL-17F[-/-] mice after *C. rodentium* infection [8], suggesting that IL-17A and IL-17F are important for induction of β-defensin expression. In our study, we showed that JMJD2D deficiency impaired the expression of IL-17F, β-defensin-1, β-defensin-3, and β-defensin-4 in the colonic epithelial cells and in CMT93 cells, indicating that JMJD2D are important for the expression of IL-17F, β-defensin-1, β-defensin-3, and β-defensin-4. IL-17F knockdown dramatically reduced the expression of β-defensin-1, β-defensin-3, and β-defensin-4 in CMT93 cells, whereas ectopic IL-17F expression or recombinant IL-17F treatment increased the expression of β-defensin-1, β-defensin-3, and β-defensin-4 in JMJD2D-knockdown CMT93 cells, suggesting that IL-17F is responsible for the expression of β-defensins and mediates the up-regulation of β-defensins by JMJD2D.

IL-17F-deficient mice are susceptibility to *C. rodentium* infection due to the defect in β-defensin production [8], indicating that IL-17F are required for the control of *C. rodentium* infection. In addition to innate immune cells, IL-17F is highly expressed in colonic epithelial cells [8]. Because our data showed that JMJD2D in nonhemopoietic cells was important for host defense against *C. rodentium* infection, so we focus on studying the expression of IL-17F in the colonic epithelial cells. When JMJD2D[-/-] mice displayed more *C. rodentium* burdens compared with wild-type mice on days 7 and 14 after *C.rodentium* infection, we observed a decreased expression of IL-17F and β-defensins in the colonic epithelial cells of JMJD2D[-/-] mice. Thus, impaired IL-17F and β-defensin production in the colonic epithelial cells contributes to the sensitivity of JMJD2D[-/-] mice to *C. rodentium*.

It has been reported that the transcription factor STAT3 in the colonic epithelial cells can be activated by *C. rodentium* infection, and intestinal epithelial STAT3 is essential for the clearance of *C. rodentium* infection [33]. Our recent study has demonstrated that JMJD2D can serve as a coactivator for STAT3 to enhance its transcriptional activity [34]. Implied by these information, when we found a STAT3 binding site on the IL-17F promoter, we hypothesized that JMJD2D may cooperate with STAT3 to induce IL-17F expression. Indeed, our current data showed that JMJD2D could be recruited to the STAT3 binding site on the IL-17F promoter and JMJD2D could cooperate with STAT3 to enhance IL-17F transcription, indicating that JMJD2D regulates IL-17F expression at the transcriptional level through activation of STAT3 signaling. In the current study, we also observed JMJD2D deficiency reduced STAT3 phosphorylation without or with *C. rodentium* infection. The possible mechanism could involve JMJD2D-mediated regulation of upstream signaling pathways that converge on STAT3 phosphorylation. Our recent study has demonstrated that JMJD2D can elevate IFNGR1 expression, which modulated STAT3 phosphorylation in CMT93 cells [34].

Our previous study has shown that JMJD2D$^{-/-}$ mice are more sensitive to dextran sulfate sodium-induced colitis (DSS) due to the defect in the colonic regeneration, which is characterized by an increased in colonic epithelial cells apoptosis and a decrease in colonic epithelial cell proliferation via impaired Hedgehog signaling activation [17], indicating that JMJD2D plays an important protective role in DSS-induced colitis. In the current study, we also observed lower body weight and shorter colon length in JMJD2D$^{-/-}$ mice compared to wild-type mice on days 7 and 14 after infection (**Fig 1B and 1E**). Considering the important role of IL-23/IL-22 axis in the initiation of immune response, especially crypt proliferation and regenerative repair, we detected the expression levels of IL-23 and IL-22 in colon and isolated intestinal epithelial cells. No significant differences were observed in the expression of IL-22 and IL-23 in the colons of wild-type and JMJD2D$^{-/-}$ mice on days 7 and 14 after *C. rodentium* infection. On day 14 after *C. rodentium* infection, JMJD2D deficiency led to impaired IL-22 expression in colonic epithelial cells. No significant difference was observed in IL-23 expression in colonic epithelial cells of wild-type and JMJD2D$^{-/-}$ mice infected with *C. rodentium* (**S4A Fig**). Additionally, we employed Ki67 staining to define the regenerative crypt hyperplasia response. The results revealed comparable numbers of Ki67$^+$ cells per crypt between wild-type mice and JMJD2D$^{-/-}$ mice without or with *C. rodentium* infection (**S4B Fig**). This might be attributed to the peak expression of IL-22 occurring on the third day after infection [7], while differences in regenerative response occurred in the early stages following infection. Furthermore, we treated wild-type and JMJD2D-knockdown CMT93 cells with IL-22 and then examined the expression of JMJD2D, p-STAT3, and Reg3β, as well as the recruitment of STAT3 on Reg3β promoter in wild-type and JMJD2D-knockdown CMT93 cells without or with IL-22 treatment. JMJD2D knockdown reduced IL-22-induced STAT3 phosphorylation and Reg3β expression, as well as reduced the recruitment of STAT3 on the Reg3β promoter (**S4C–S4E Fig**), suggesting that JMJD2D may play a crucial regulatory role in the IL-22-STAT3-Reg3β-mediated damage repair, and the specific role and molecular mechanisms remain to be elucidated, and we propose this as a new topic for future investigation. Our current study mainly focused on the protective role of JMJD2D in *C. rodentium* infection via increasing β-defensin production by upregulation of IL-17F expression. Therefore, our findings suggest that JMJD2D plays essential protective roles in intestine in response to different stimulations via different mechanisms.

Based on the current and previous studies, we propose a working model for the role of JMJD2D in protecting colonic epithelial cells from *C. rodentium* infection: *C. rodentium* infection activates NF-κB signaling to induce JMJD2D expression; on one hand, JMJD2D activates STAT3 signaling by enhance STAT3 phosphorylation; on the other hand, JMJD2D cooperates with STAT3 to promote IL-17F expression to further induce the expression of antimicrobial

**Fig 7. Schematic model for the role of JMJD2D in protecting colonic epithelial cells from *C. rodentium* infection.** *C. rodentium* infection activates NF-κB signaling to induce JMJD2D expression; on one hand, JMJD2D activates STAT3 signaling by enhance STAT3 phosphorylation; on the other hand, JMJD2D cooperates with STAT3 to promote IL-17F expression to further induce the expression of antimicrobial peptide β-defensins to eliminate *C. rodentium* by interacting with STAT3 and being recruited to the IL-17F promoter to demethylate H3K9me3.

peptide β-defensins to eliminate *C. rodentium* by interacting with STAT3 and being recruited to the IL-17F promoter to demethylate H3K9me3 (**Fig 7**).

## Supporting information

**S1 Fig.** The efficiency of bone marrow transplantation was assessed by designing distinct probes targeting both wild-type and knockout alleles of JMJD2D in murine peripheral blood genomic DNA via digital PCR. n = 6.
(TIF)

**S2 Fig.** There is no significant difference in the expression of Reg 3β, Reg 3γ, CRAMP, RELM-β, S100A8, S100A9, and Muc5Ac in the colonic epithelial cells of wild-type and JMJD2D$^{-/-}$ mice after *C. rodentium* infection. Results are representative of three independent experiments.
(TIF)

**S3 Fig.** (**A**) Heat-killed *C. rodentium* treatment could activate NF-κB signaling by increased phosphorylation of p65. (**B**) STAT3 phosphorylation was impaired in the colonic epithelial cells of JMJD2D$^{-/-}$ mice after *C. rodentium* infection.
(TIF)

**S4 Fig.** (**A**) No significant differences were observed in the expression of IL-22 and IL-23 in the colons of wild-type and JMJD2D$^{-/-}$ mice on days 7 and 14 after *C. rodentium* infection. On day 14 after *C. rodentium* infection, JMJD2D deficiency led to impaired IL-22 expression in colonic epithelial cells. No significant difference was observed in IL-23 expression in colonic epithelial cells of wild-type and JMJD2D$^{-/-}$ mice infected with *C. rodentium*. (**B**) Ki67 staining revealed comparable numbers of Ki67$^+$ cells per crypt between wild-type mice and JMJD2D$^{-/-}$ mice without or with *C. rodentium* infection. (**C, D**) JMJD2D knockdown reduced IL-22-induced STAT3 phosphorylation (**C**) and Reg3β expression and (**D**). (**E**) JMJD2D knockdown reduced the recruitment of STAT3 on the Reg3β promoter without or with IL-22 treatment. Results are representative of three independent experiments. **$P<0.01$; ***$P<0.001$. (TIF)

**S1 Data. Excel spreadsheet containing, in separate sheets, the underlying numerical data and statistical analysis for Figure panels.**
(XLSX)

## Author Contributions

**Data curation:** Yong Zhang, Bei Li, Yilin Hong.

**Formal analysis:** Yong Zhang.

**Funding acquisition:** Zaifa Hong, Xiaochun Xia, Chundong Yu, Wenbo Chen.

**Investigation:** Yong Zhang, Yilin Hong, Ping Luo, Pingli Mo.

**Methodology:** Yong Zhang, Bei Li.

**Resources:** Chundong Yu, Wenbo Chen.

**Supervision:** Chundong Yu, Wenbo Chen.

**Validation:** Bei Li.

**Visualization:** Yong Zhang.

**Writing – original draft:** Yong Zhang.

**Writing – review & editing:** Bei Li, Yilin Hong, Zaifa Hong, Xiaochun Xia, Pingli Mo, Chundong Yu, Wenbo Chen.

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
