## [Decision Letter · Decision Letter 0]

13 Mar 2024

Dear Dr. Yu,

Thank you very much for submitting your manuscript "Histone demethylase JMJD2D protects against enteric bacterial infection via up-regulating colonic IL-17F to induce β-defensin expression" for consideration at PLOS Pathogens. As with all papers reviewed by the journal, your manuscript was reviewed by members of the editorial board and by several independent reviewers. In light of the reviews (below this email), we would like to invite the resubmission of a significantly-revised version that takes into account the reviewers' comments.

We cannot make any decision about publication until we have seen the revised manuscript and your response to the reviewers' comments. Your revised manuscript is also likely to be sent to reviewers for further evaluation.

Sincerely,

Jose Luis Balcazar, Ph.D.

Academic Editor

PLOS Pathogens

David Skurnik

Section Editor

PLOS Pathogens

Michael Malim

Editor-in-Chief

PLOS Pathogens

orcid.org/0000-0002-7699-2064

Reviewer's Responses to Questions

**Part I - Summary**

Reviewer #1: In this new article, Zhang et al identified a new role of JMJD2D in colonic epithelial cells to help defend against C. rodentium infection. They then demonstrate that JMJD2D in colonic epithelial cells increases STAT3 activity to work together for up-regulating IL-17F, thereby increasing beta-defensin expression.

The experimental design is logic and well considered. The results are clear and strong to support the major conclusions, which is of interest to the related research fields.

Reviewer #2: How the gene regulatory machinery, including histone modifying enzymes, are involved in innate immunity processes is an important area of research with many implications. In the study by Yong Zhang et al, the team address the role of H3K9m2/3 demethylase JMJD2D (also known as KDM4D) in this process. They focus on the Citrobacter rodentium infection model in the mouse that bears relevance to understanding infection in human, too, e.g. infection by pathogenic E. coli. This study complements previous work by the team that examined the role of JMJD2D in experimental colitis in the mouse.

In this paper the authors show that deletion of JMJD2D causes increased pathology on Citrobacter rodentium infection (Fig. 1) and indicate a role of JMJD2D in non-bone marrow-derived cells in this protective function, presumably the intestinal epithelium (Fig.2). They implicate a role of JMJD2D in regulating specific anti-bacterial β-defensins and IL-17F expression in colonic epithelial cells in this protective function (Fig.3). They show in cells in culture that IL-17F is required for the expression of several defensins (Fig.4). Importantly, ectopic IL-17F expression in JMJD2D–knockdown cells or treating these cells with recombinant IL-17F rescued the expression of β-defensin-1, β-defensin-3, and β-defensin-4. The authors provide evidence that heat-killed C. rodentium treatment of cells in culture causes JMJD2D to cooperate with STAT3 to induce IL-17F expression by demethylating H3K9me3 on the IL-17F promoter (Fig.5). They provide data supporting that the demethylase activity of JMJD2D is required for the induction of IL-17F and β-defensins (Fig.6).

This is a fairly clear-cut and interesting study that ads to the picture how histone modifying factors are involved in innate immunity processes controlling infection. The data are convincing, the presentation is logic and I see now major issue.

Reviewer #3: This manuscript nicely describes work on how a Histone Demethylase (JMJD2D) is induced upon infection/inflammation (not necessarily novel), and that it likely plays a role in controlling epithelial IL-17F, which the authors suggest is due to controlling phospho-Stat3 and/or Stat3-JMJD2D interaction near the IL-17F locus. Subsequently, the IL17F controls B-Defensin expression, which would act as antimicrobials to control a bacterial infection. Together, the authors propose JMJD2D is required for the demethylation of H3K9 to allow for gene expression (Il17F) activation. While both the in vitro and in vivo data is strong and nicely presented, there are a few gaps that are not addressed.

Strengths: Clear phenotype in the KO during Citro – weight loss and bacterial burden are clearly affected, which correlates with the histology. In addition, Figs 5&6 nicely show some mechanistic work that is quite strong.

Weakenesess: the bone marrow exps lack the data that shows that the bone marrow transplant was successful (and what percentage). In addition, the biological data in the BM experiments is less convincing than the full KO. The proposed mechanisms (reduced pStat3) should be evaluated in the in vivo exps. Normally, I would expect reduced pSTAT3 to also lead to reduced Reg3b/g gene expression (classical pSTAT3 target genes), which is not the case. Finally, classically, it is thought that IL-23-IL-22 axis is responsible for the initial innate response to Citro (this induces pStat3 in epithelium) – the authors do not place their findings into this context and only relate it to IL-17A. This is important as the authors see clear effects such as weightloss at the initial stage.

**Part II – Major Issues: Key Experiments Required for Acceptance**

Reviewer #1: n/a

Reviewer #2: (No Response)

Reviewer #3: To me there are 3 aspects that needs to be addressed:

- Confirmation of how successful the BM transplant was.

- There needs to be a distinction between the ‘regenerative’ phenotype from the previous paper and the antimicrobial B-defensin phenotype proposed here in this manuscript (i.e. the impaired regeneration could also drive the inflammation/citro phenotype). What happens to the initial IL-23/IL-22 axis in these animals? IL-22 is thought to induce crypt hyperplasia (regenerative response), is this impaired (and this is causing the phenotype?)? To test, IL23/IL22 can be measured by qPCR, and Ki67 stain can define regenerative crypt hyperplasia response.

- The proposed mechanism is via STAT3 – however, this needs to be verified in vivo: What are pSTAT3 levels in vivo after infection. As other pSTAT3 targets (Reg3b/g) are not altered in the KO, why/how is IL17F specifically affected/tuned. pSTAT3 staining or by western should be performed (of infected animals to determine the response) to determine that this is the mechanism by which IL17F is controlled in vivo after infection.

**Part III – Minor Issues: Editorial and Data Presentation Modifications**

Reviewer #1: 1. Figure 3G, it is IL-17F protein level , not mRNA level.

2. Also confirm the IL-17Ra and IL-17RC are mRNA levels or protein levels.

3. Reference #35 miss the journal name. Authors need to go over carefully all the references to make sure the information is intact.

4. It is unclear how JMJD2D increases STAT3 phosphorylation.

Reviewer #2: Given that the authors performed only targeted gene expression and chromatin analysis, they should tone down the discussion a bit when they write (line 487 onwards): “ Thus, higher C. rodentium burdens in JMJD2D-/- mice are caused by reduced expression of β-defensin-1, β-defensin-3, and β-defensin-4 after C. rodentium infection, which lead to more severe colonic injury ” by rephrasing, e.g., “Thus, we propose that higher C. rodentium burdens in JMJD2D-/- mice are caused by a reduced innate immunity response, such as expression of β-defensin-1, β-defensin-3, and β-defensin-4 after C. rodentium infection, which lead to more severe colonic injury” or similar. There may be additional protective pathways that depend on JMJD2D that the authors simply have not screened for.

Minor comments:

Line 33, abstract: “The enteric pathogen Citrobacter rodentium (C. rodentium) model was used to mimics clinical colonic infection.” Should be “to mimic”…

Line 101: “IL-17F.(11)”, the reference should come before the period/full stop (.).

Reviewer #3: (No Response)

PLOS authors have the option to publish the peer review history of their article (what does this mean?). If published, this will include your full peer review and any attached files.

Reviewer #1: **Yes: **Huang, Wendong

Reviewer #2: **Yes: **Patrick Varga-Weisz

Reviewer #3: No
---

## [Decision Letter · Decision Letter 1]

5 Jun 2024

Dear Dr. Yu,

We are pleased to inform you that your manuscript 'Histone demethylase JMJD2D protects against enteric bacterial infection via up-regulating colonic IL-17F to induce β-defensin expression' has been provisionally accepted for publication in PLOS Pathogens.

Best regards,

Jose Luis Balcazar, Ph.D.

Academic Editor

PLOS Pathogens

David Skurnik

Section Editor

PLOS Pathogens

Michael Malim

Editor-in-Chief

PLOS Pathogens

orcid.org/0000-0002-7699-2064

The authors have addressed all reviewers' comments and made substantial improvements to the manuscript. Many thanks!

Reviewer Comments (if any, and for reference):

Reviewer's Responses to Questions

**Part I - Summary**

Reviewer #1: The authors have addressed the previous comments. It is now acceptable for publication in PLOS Pathogens.

Reviewer #2: The authors have carefully addressed all reviewers' comments and improved the manuscript.

Reviewer #3: The authors have addressed all my issues. Compliments on the great work!

**Part II – Major Issues: Key Experiments Required for Acceptance**

Reviewer #1: (No Response)

Reviewer #2: N/A

Reviewer #3: (No Response)

**Part III – Minor Issues: Editorial and Data Presentation Modifications**

Reviewer #1: (No Response)

Reviewer #2: N/A

Reviewer #3: (No Response)

PLOS authors have the option to publish the peer review history of their article (what does this mean?). If published, this will include your full peer review and any attached files.

Reviewer #1: **Yes: **Wendong Huang

Reviewer #2: No

Reviewer #3: No

---

## [Editor Report · Acceptance letter]

12 Jun 2024

Dear Dr. Yu,

We are delighted to inform you that your manuscript, "Histone demethylase JMJD2D protects against enteric bacterial infection via up-regulating colonic IL-17F to induce β-defensin expression," has been formally accepted for publication in PLOS Pathogens.

Best regards,

Michael Malim

Editor-in-Chief

PLOS Pathogens

orcid.org/0000-0002-7699-2064